# Time- and Dose-Dependent Effects of Dietary Deoxynivalenol (DON) in Rainbow Trout (*Oncorhynchus mykiss*) at Organism and Tissue Level

**DOI:** 10.3390/toxins14110810

**Published:** 2022-11-20

**Authors:** Paraskevi Koletsi, Geert F. Wiegertjes, Elisabeth A. M. Graat, Philip Lyons, Johan Schrama

**Affiliations:** 1Aquaculture and Fisheries Group, Wageningen University and Research, 6708 WD Wageningen, The Netherlands; 2Adaptation Physiology Group, Wageningen University and Research, 6708 WD Wageningen, The Netherlands; 3Alltech Biotechnology Inc., A86 X006 Dunboyne, Ireland

**Keywords:** mycotoxins, aquaculture, growth, protein retention, histology, liver, intestine, gene expression

## Abstract

This study with juvenile rainbow trout evaluated the effects of dietary exposure to deoxynivalenol (DON) at industrially relevant doses (up to 1.6 mg/kg) on growth performance, the liver, and the gastrointestinal tract. Fifteen groups of 30 fish each were given one of five dietary treatments in triplicate: (1) control diet (CON; DON < 100 µg/kg feed), (2) naturally DON-contaminated diet (ND1) with a DON content of 700 µg/kg in the feed, (3) ND2 with a DON content of 1200 µg/kg feed, (4) a pure DON-contaminated diet (PD1) with 800 µg/kg of DON in the feed, and (5) PD2 with DON at a concentration of 1600 µg/kg in the feed. The feeding trial lasted eight weeks: six weeks of restrictive feeding followed by two weeks of *ad libitum* feeding. Exposure to DON during restrictive feeding for six weeks did not affect the growth performance of trout but did lead to a reduction in retained protein in fish fed with higher doses of DON in the ND2 and PD2 groups. During the two following weeks of *ad libitum* feeding, feed intake was similar among all groups, but body weight gain was lower in the ND2 and PD2 groups and feed efficiency was higher in PD2 (week 8). Histopathological assessment revealed liver damage, including altered nuclear characteristics and haemorrhages, in groups fed higher doses of natural DON (ND2) after just one week of restrictive feeding. Liver damage (necrosis and haemorrhage presence in ND2) was alleviated over time (week 6) but was again aggravated after *ad libitum* exposure (week 8). In contrast, gastrointestinal tract damage was generally mild with only a few histopathological alterations, and the absence of an inflammatory cytokine response was demonstrated by PCR at week 8. In conclusion, *ad libitum* dietary exposure of rainbow trout to either natural or pure DON resulted in reduced growth (dose-dependent), while restrictive exposure revealed time-dependent effects of natural DON in terms of liver damage.

## 1. Introduction

Mycotoxins have been well-documented as frequent natural contaminants in aquafeeds in Europe, where deoxynivalenol (DON) is the most prevalent toxin [1,2,3,4,5]. The risk of contamination has been highlighted in the global aquaculture sector and is becoming increasingly relevant given the increased use of plant-derived ingredients in fish feeds [1,2,3,4,5,6,7,8,9]. Plant-based raw materials can have a high nutritional value within diets for farmed fish but conversely represent potential substrates for the growth of mycotoxin-producing fungi [10]. Climate change forecasts suggest that climate-driven fungal growth factors could exacerbate the potential for mycotoxin production in crops and increase the risk of contamination levels in animal feeds [11,12,13]. Therefore, research into the potential effects of diets contaminated with mycotoxins such as DON on fish performance and health is timely.

DON is produced by *Fusarium* fungi that contaminate crops prior to harvest, and at this stage, it is difficult to apply prevention strategies [14]. Additionally, DON is a heat-stable toxin, meaning that high temperatures during feed extrusion cannot eliminate DON and guarantee its absence from the final fish feeds [15]. Rainbow trout is amongst the most sensitive species to this toxin [1]. In rainbow trout, exposure to DON (≥800 µg/kg) under *ad libitum* feeding affects fish in a manner similar to terrestrial animals and is manifested through reduced feed intake and weight gain [16,17,18,19,20,21,22]. Moreover, DON suppresses the retention efficiency of dietary nitrogen and energy in rainbow trout at doses ≥1300 µg/kg [16,18,19,21,22]. Histopathological investigations have described liver damage in trout after exposure to DON doses ≥1400 µg/kg and revealed congestion and subcapsular edema with a fibrinous network, fatty infiltration, phenotypically altered hepatocytes [16], vacuolation of hepatocytes, necrosis, scattered haemorrhages [20], and a decrease in the number of mitotic cells [21]. Thus far, histopathological assessments have primarily been conducted via qualitative or semi-quantitative methods [23], meaning that quantitative studies of potential damage to the liver and/or intestine caused by DON are largely unreported, especially in rainbow trout.

Effects of dietary exposure to DON can be studied using feed ingredients naturally contaminated with DON (“natural” DON) to compose experimental diets or through the addition of pure DON to diets. Feed ingredients with “natural” DON often also contain other types of mycotoxins [16,18,21,22,24], whereas using pure DON excludes co-exposure to other toxins. In a direct comparison between natural and pure DON (at a dose of 2100 µg/kg) in rainbow trout, no differences in growth performance or nutrient utiliaation efficiency metrics were observed [21]. In contrast, our meta-analysis indicated that natural DON exposure resulted in a more pronounced reduction in feed intake and growth of rainbow trout than exposure to pure DON [1]. Time-dependent effects remain somewhat understudied in rainbow trout. So far, only one study has assessed the effects of exposure time to natural or pure DON-contaminated diets on rainbow trout, showing time-related histopathological changes in both the pyloric caeca and the liver [21]. Only in common carp have researchers more specifically addressed the time effects of DON (953 μg/kg) (7, 14, 26, and 56 days) [25,26]. Liver damage and a reduction in the activity of specific biotransformation enzymes were present only until day 26 [26], while up-regulation of pro- and anti-inflammatory cytokine gene expression in the spleen, liver, and intestine was found only at day 14 [25]. 

A critical aspect of evaluating the effects of dietary exposure to DON on growth is the choice of feeding regime: restrictive versus *ad libitum*. Our meta-analysis on rainbow trout, which only assessed studies with *ad libitum* exposure, showed reduced growth as an outcome of reduced consumption of DON-contaminated feed [1]. If DON exposure disturbs appetite, experimental designs using *ad libitum* feeding would generate differences in feed consumption among experimental treatments and thus generate differences in DON intake. Therefore, *ad libitum* feeding designs do not allow for measuring the direct effects of DON on growth performance. Restrictive feeding experiments should thus be more informative for direct DON-related effects on growth performance. Only pair-fed treatments added to *ad libitum* feeding experiments have shown either direct effects of DON on growth performance [16] or no effect [17]. Well-designed studies with dietary exposure of rainbow trout to DON to examine its direct effects on growth are rare.

The purpose of the present study was to elucidate the direct effects of two different types of DON (natural versus pure) on both growth performance and health metrics of rainbow trout (*Oncorhynchus mykiss*) via a detailed histopathological examination of the liver and gastrointestinal tract. Firstly, the experimental design of the study was based on restrictive feeding for six weeks to ensure equal feed intake, with the aim to reveal direct effects of DON rather than indirect effects caused by differences in feed intake. Secondly, the inclusion of an early sampling point at one week permitted an investigation of whether DON-induced effects would change over time between week 1 and week 6. Thirdly, the inclusion of a final experimental period of two weeks of *ad libitum* feeding allowed for the assessment of indirect effects of DON caused by a reduction in feed intake. Finally, we studied the observed dose-dependent reduction of performance parameters and time-dependent liver damage after dietary exposure of rainbow trout to DON.

## 2. Results

### 2.1. Performance 

No mortality, notable differences in feed acceptance, or abnormal behavioural responses were noted during the experiment. Performance parameters were not significantly different between diets during the restrictive feeding period of six weeks, except in the case of retained protein and protein retention efficiency (*p* < 0.01; Table 1). Rainbow trout fed diets ND2 and PD2 retained less protein and had a lower protein retention efficiency compared to trout fed the control (CON) diet. Trout fed the PD1 diet were similar to trout on the CON diet but differed from fish fed the PD2 diet regarding protein gain and protein retention efficiency.

During the subsequent ad libitum feeding period of two weeks, feed intake, HSI (a indicator of relative liver size), and condition factor K did not differ between control and DON-contaminated diets (Table 2). However, absolute growth (g/d) and specific growth rate (SGR, % BW/d) were reduced in rainbow trout fed the diets containing the highest DON contamination level (ND2 and PD2), compared to those fed the CON diet, via an *ad libitum* feeding regime (*p* ≤ 0.01; Table 2). FCR was only increased for fish fed the PD2 diet compared to the CON-fed fish (*p* ≤ 0.05; Table 2).

### 2.2. Health

#### 2.2.1. Histopathological Assessment of the Liver

Qualitative observations suggested that liver cells of fish fed the DON-contaminated PD2 diet had a lower degree of glycogen vacuolization (lower degree of pink coloration in PAS-stained hepatocytes; Figure 1) compared to fish fed the CON diet. The quantitative assessment of the glycogen vacuolization score of hepatocytes revealed an interaction effect between diet and time (Table 3; *p* ≤ 0.05); during the restrictive feeding period, while glycogen vacuolization was similar among dietary treatments and remained unaltered over time. However, at the end of the *ad libitum* feeding period, fish fed the PD2 diet had reduced hepatic glycogen vacuolization. Lipid vacuolization was unaffected by diet and did not alter with time. No differences in the lipid vacuolization of liver cells, identified as white spherical droplets in the hepatocytes, were observed (Figure 1).

Only a diet effect was present in the models of nuclei characteristics, pyknosis, and pleomorphism (see Figure 1 for qualitative indication), and it did not change over time. Averaged over all sampling moments, trout fed ND2 and PD2 diets had an increased level of pyknotic and pleomorphic nuclei, i.e., altered cells, than trout fed a CON diet, without differences between natural (ND2) and pure DON (PD2). During *ad libitum* feeding, only fish fed the ND2 diet had a higher occurrence of pyknotic and pleomorphic nuclei (week 8), indicating that overexposure to natural DON altered the nuclei of cells.

Further qualitative observations of additional pathological signs, including necrosis (recognized as disrupted cell structure), haemorrhage (recognized as accumulated red blood cells outside blood vessels), and inflammation (recognized as accumulated leucocytes), suggested an early and severe effect of ND2 and PD2 at week 1 on liver health (Figure 1). This effect seemed to be time-dependent, since at week 6 of restrictive feeding, most of the pathological signs were no longer different from the control, except for some signs of inflammation in the ND2 group. After two weeks of *ad libitum* feeding (week 8), pathological signs seemed to reappear in the livers of trout fed ND2 (signs of necrosis and inflammation) and PD2 (signs of necrosis and haemorrhage) diets (Figure 1). Indeed, quantitative assessment (Table 3) confirmed an interaction effect between diet and time for the presence of necrosis (*p* ≤ 0.01), necrosis score (*p* ≤ 0.05), haemorrhage (*p* ≤ 0.001) and inflammation (*p* ≤ 0.05). Only for fish fed the ND2 diet, the presence of necrosis and haemorrhage decreased while inflammation increased over time (from week 1 to week 6).

For all liver parameters, diet, time, and their interaction explained a rather low proportion of the variance. The ten measurements per fish for the liver parameters are assumed not to be independent. This is shown by the high proportion of unexplained variation due to the effect of fish (>0.5) in pyknosis and pleomorphism (Table 3). This effect was moderate for necrosis (22%) and inflammation (16%) and only 4% for haemorrhage.

#### 2.2.2. Histopathological Assessment of the Gastrointestinal Tract

All measured parameters, indicators for mucosal health, and morphology are listed in Appendix A. The thickness of the sub-epithelial mucosa, mucosal fold height, the width of the mucosal fold, lamina propria, stratum granulosum, enterocytes, and muscularis width all increased over time, being highest at week 8 (Appendix A). The interaction effects between diet and time were only significant for mucosal fold height in the pyloric caeca (Figure 2a; *p* ≤ 0.01), mucosal fold width in the hindgut (Figure 2b; *p* ≤ 0.01) and enterocyte width in the hindgut (Figure 2c; *p* ≤ 0.05). In the pyloric caeca, the height of the mucosal fold changed over time only within fish fed the ND2 diet. Specifically, mucosal fold height of ND2 in week 8 is significantly different from week 6, and also from the control diet at all weeks and from weeks 1 and 8 in the case of PD2. In the hindgut, a time effect was detected only for mucosal fold and enterocyte metrics in the PD2 diet, where in both cases the widths were significantly higher in week 8 compared to week 1. For the majority of the other histological parameters investigated in the gastrointestinal tract, there were no indications of an effect of diet or an interaction between diet and time (Appendix A). 

### 2.3. Assessment of Inflammation by Gene Expression

In order to examine the putative effects of DON on intestinal health in more detail, expression analyses of inflammatory genes were performed by PCR. Gene expression patterns of selected pro-inflammatory cytokines (IL-1*β*, IL-8, and TNF-*α*) were measured in the pyloric caeca and hindgut of rainbow trout at the end of the *ad libitum* feeding period, week 8, where suspected damage would be greatest. Exposure to PD2 resulted in a down-regulation of IL-1*β* in the pyloric caeca (Figure 3a), but not an up-regulation. In the same tissue, relative expression of IL-8 and TNF-*α* showed a trend of down-regulation in the PD2 group, although this was not statistically significant. Furthermore, in the ND2 group, gene expression of pro-inflammatory cytokines was also not significantly up- or down-regulated. Gene expression analysis in the hindgut (Figure 3b) showed no effect of dietary treatment. Taken together, gene expression analysis confirmed the absence of strong effects of DON in the gastrointestinal tract.

## 3. Discussion

The purpose of this study was to elucidate the effects of DON on growth performance and on the gut and liver health of rainbow trout. DON was fed at industrially relevant doses of up to 1.6 mg/kg and derived as pure DON or natural DON for inclusion in experimental diets. The study focused on measuring the direct effects of restricted feeding on growth performance and the quantification of histopathological effects in the liver and gastrointestinal tract. Here, we discuss the dose-dependent reduction of growth performance due to pure/natural DON and the time-dependent effects of natural DON on the liver of rainbow trout.

### 3.1. Performance

In the current study, dietary DON inclusion levels of ≥1200 µg/kg had some effect on performance during restricted feeding, but such effects were more pronounced after *ad libitum* feeding. A critical aspect of evaluating the effects of dietary DON exposure on performance is the choice of feeding regime: restrictive versus *ad libitum*. Restrictive feeding should reveal the direct effects of a fixed, daily DON intake, while *ad libitum* feeding should reveal the direct effects of DON plus putative indirect effects on impaired feed intake. Our restrictive feeding experiment induced reductions in retained protein and protein retention efficiency, but only in experimental treatments fed diets containing the highest level of DON contamination (ND2: 1200 µg/kg and PD2: 1600 µg/kg). This reduction is in agreement with an earlier study in trout, although a higher dose of DON (2600 µg/kg) was used [16]. Our findings indicate that DON may directly inhibit protein synthesis and/or increase maintenance requirements, which could induce increased protein catabolism and impaired nutrient utilization.

*Ad libitum* feeding reduced body weight gain in the ND2 and PD2 diets and feed efficiency in PD2. Restrictive feeding did not induce strong effects on growth performance, despite the above-mentioned reduction in retained protein in fish fed with higher doses of DON. The lack of DON-related effects on trout growth during restrictive feeding may be explained by the absolute amount of DON ingested, calculated as estimated daily intake (EDI; µg/g BW/day). There is likely a threshold concentration of DON that individual animals can tolerate, below which no adverse effects would be recorded, which could help explain the stronger effect seen after the *ad libitum* feeding. Indeed, during restrictive feeding, the EDI for trout, which received the highest dose (PD2; 1600 µg/kg), was 0.038 μg DON/g BW/day, while during *ad libitum* feeding the EDI was as high as 0.054 μg DON/g BW/day (Appendix A). Furthermore, the biomass measurements at weeks six and three (no reported data) showed no effects of DON, and it is rational to assume that there is no adaptation over time in terms of growth. In future investigations, it may be informative to include an early (week 1) sampling point to determine direct effects of DON on growth performance, similar to observations in pigs [27,28]. Alternatively, longer periods of restrictive feeding up to eight weeks of exposure might reveal more prominent direct effects of DON on rainbow trout growth in future studies. 

*Ad libitum* feeding did not induce differences in feed intake, despite the above-mentioned reduced body weight gain and feed efficiency measurements noted in the ND2 and PD2 groups, whereas other studies did report reduced feed intake [16,17,18,19,20,21,22,24,29]. Also, in our meta-analysis, we predicted that each additional mg/kg of DON in trout feed would lead to an exponential decline in feed intake, with a rate of 18.8% [1]. A possible explanation for the absence of DON-induced feed refusal in the current experiment might be the relatively short duration of two weeks of *ad libitum* exposure. The majority of other studies maximum exposure times of up to eight weeks. Another hypothesis could be derived from the restrictive feeding period of six weeks, prior to the *ad libitum* feeding, during which fish may have adapted to tolerate the consumption of DON-contaminated feeds. 

Feed ingredients naturally contaminated with DON often also contain other types of mycotoxins, whereas the use of DON produced under controlled laboratory conditions excludes cross-contamination [1]. In our study, a direct and fair comparison between natural (ND) and pure DON (PD) proved difficult because DON levels in naturally contaminated diets were lower than anticipated. After restrictive feeding, performance was not different between ND2 and PD2. Yet, after *ad libitum* feeding, ND2 (at 1200 μg/kg) had a stronger effect on feed intake (but not significant) than PD2 at 1600 μg/kg. Our observations therefore are not in disagreement with the results of our meta-analysis for trout [1], which suggested the presence of combined effects due to the potential co-contamination of natural sources of DON with other *Fusarium* toxins.

### 3.2. Liver

The liver is the most studied organ evaluated for DON toxicity in fish species, although there are inconsistent approaches to scoring (semi-quantitative/qualitative) and staining protocols (H&E/PAS) among studies screening for the same pathological indicators in histological assessments [23]. In the present study, a semi-quantitative approach was used to evaluate histopathological parameters in the liver, meeting the requirements of fundamental concepts for the semi-quantitative scoring of tissues [30]. A detailed scoring protocol was developed that translated qualitative information from microscopic liver images into data suitable for statistical analyses. This protocol quantified the severity of DON toxicity in trout liver using histopathological parameters and is the first study in trout to measure the toxic effects of DON over time within a restrictive feeding period.

Restrictive DON feeding did not appear to affect glycogen or lipid vacuolization. During restrictive feeding (weeks 1–6), time-related DON effects were noted in the form of haemorrhage, inflammation, and necrosis. Notably, these changes over time were found only after feeding the ND2 diet, implying a more severe toxic response using co-contaminated feed. Furthermore, we identified numerical but not statistically significant differences for the presence of pyknosis and pleomorphism in nuclei in week 6 (higher presence in DON-contaminated diets than the CON diet), most likely due to the low sample size and consequently lower statistical power. Studies comparable to ours have only been performed in common carp, where restrictive exposure to 953 µg/kg DON for six weeks did not induce differences in hepatic glycogen vacuolization, but did increase lipid accumulation [31]. Carp showed histopathological indicators of impaired liver functioning on days 14 (hyperaemia, vacuolization, and dilation of sinusoids) and 26 (fat aggregation and dilation of sinusoids), but nothing indicated histopathological liver damage at the end of the experiment (day 56). Based on these common findings, at least in situations of restrictive exposure, it is reasonable to assume early acute responses to DON toxicity could possibly be diminished by physiological adaptation mechanisms. 

*Ad libitum* feeding of DON (weeks 6–8) affected almost all histopathological parameters scored within the liver; glycogen vacuolization, pyknosis, pleomorphism of nuclei, increased presence of necrosis, and haemorrhages. So far, studies in trout [20,21] have not made a distinction between lipid- and glycogen-type vacuolization. At present, although the mechanism of glycogen depletion in hepatocytes of DON-treated fish is not yet fully understood, glycogen vacuolization appears to be a good indicator of DON-induced liver effects. Other experiments in trout using high doses of DON (2700 mg/kg) reported multiple areas of necrosis with scattered haemorrhages [20] and phenotypically altered hepatocyte nuclei (pyknosis and karyolysis) in trout fed 2600 µg/kg natural DON [16]. The presence of necrosis in the livers of our control-treated fish, although the percentages showed a declining trend along the sampling points, suggests an adaptation to the high-carbohydrate diet. Another points is that we observed dislocated nuclei at the edges of the cells in all sampling points and treatments (data not shown), which is most likely due to high lipid vacuolization and is not necessarily a pathological finding. However, we did not find differences in hepatic lipid vacuolization as was hypothesized; DON would inhibit the lipoprotein synthesis and cause fat accumulation as droplets in the hepatocytes [32]. Perhaps the high carbohydrate content in our diets was the main factor that caused fat accumulation in the hepatocytes of all treatments, including the CON, and therefore I was not possible to detect the DON effects. An early study in trout measured hepatic lipid accumulation after six weeks of DON exposure (2600 µg/kg), although the histological appraisal in that study was conducted through H and E staining, thus rendering a distinction between glycogen and lipid droplets more challenging [16]. 

Pure DON (PD2 diet) induced a stronger reduction of glycogen vacuolization than natural DON (ND2) and affected necrosis (presence and score at week 8) more than natural DON, but both DON diets had a comparable effect on aggravating haemorrhage presence. The differences between pure and natural DON on the severity of liver damage may have been masked to some extent by the experimental diets because the DON level in the naturally contaminated diet (ND2) was slightly lower than anticipated. As a result, PD2 groups received a higher absolute DON intake during ad libitum feeding (PD2 diet: 0.054 μg/g BW/day) than ND2 groups (0.039 μg/g BW/day). Thereby, ad libitum access to the feed maximised the absolute DON intake. Eventually, DON causes injury to fish livers through, for example, lipid peroxidation induced by oxidative stress, causing necrotic tissues and pro-inflammatory responses to haemorrhages, at least in common carp [26,31]. The mechanisms at work in rainbow trout exposed to DON are yet to be fully unravelled. 

### 3.3. Gastrointestinal Tract

The effects of DON on the gastrointestinal tract (pyloric caeca, midgut, and hindgut), if any, were generally mild. DON did not affect intestinal integrity and morphology during restrictive feeding (measured at weeks 1 and 6), and only a few mild alterations were observed after the *ad libitum* exposure. At week 8, the pyloric caeca of trout fed the ND2 diet showed increased mucosal fold heights, possibly an adaptation to a DON-driven impairment in nutrient absorption. The effects were only present in the ND2 and not the PD2 diet, possibly due to the presence of multiple mycotoxins in the naturally contaminated DON sources. The pyloric caeca comprises the first part of the gastrointestinal tract and appears more sensitive to mycotoxins like DON than the midgut and hindgut. Other studies reported an increased number of dead (apoptotic/necrotic) cells and a reduced number of mitotic cells in the pyloric caeca of trout fed DON at 5900 µg/kg [21] and a reduction in the expression of tight junction proteins (TJPS) in salmon fed DON at 5500 µg/kg [33]. Next to the observations in the pyloric caeca, also in the hindgut, some relatively minor effects were seen after *ad libitum* exposure at week 8 compared to week 1, including increased widths of mucosal folds in the ND2 and PD2 groups and increased enterocyte widths for the PD2 group. Similar compensatory morphological changes, i.e., an increase in villus height in the jejunum and ileum, have also been reported for the intestine of chickens exposed to DON [34]. Possibly, the general absence of clear pathology might be attributed to the rapid absorption of DON in the upper part of the gastrointestinal tract. A toxicokinetic study in salmon showed that DON reached a peak concentration in the liver one hour after the last feeding and then decreased with a half-life (t1/2) of 6.2 h [35]. This suggests that toxicokinetic studies into the breakdown of DON in the gastrointestinal tract of rainbow trout would be of immediate relevance.

The absence of clear effects of DON on intestinal health was further detailed by gene expression analysis of the pro-inflammatory cytokines IL-1*β*, IL-8, and TNF-*α* in the pyloric caeca and hindgut. PCR analysis was performed at the end of the *ad libitum* feeding period, at week 8, where, based on the histopathological findings, the suspected damage would be greatest. If anything, cytokine gene expression in the pyloric caeca was downreglated rather than upregulated. Moreover, there was no clear pattern in the regulation of the expression of cytokine genes in the hindgut. Overall, gene expression analysis confirmed the absence of strong effects of DON in the gastrointestinal tract.

Thus far, the effects of DON on pro-inflammatory gene expression in trout appear to have been limited to studies of the spleen and head kidney [36]. Although these authors reported an up-regulation of the pro-inflammatory cytokine TNF-*α* in the head kidney of DON-fed trout (1964 μg/kg) after 23 days of exposure, the comparison between a systemic organ like the head kidney and the gastrointestinal tract is difficult to make. In Atlantic salmon [33], comparable to our findings, DON exposure (5500 µg/kg) for 8 weeks also did not induce differences in the gene expression of ILl-1β in pyloric caeca, midgut and hindgut. Most information on the effects of DON on gene expression in the gut comes from a study in carp [25]. Here, study of the proximal intestine of DON-fed carp (953 μg/kg) showed an up-regulation of pro-inflammatory (IFN-γ, TNF-α2, INOS) and also anti-inflammatory cytokines (IL-10, Arg1, and Arg2), after 14 days of exposure, returning to control levels at later time points (days 26 and 56). Taken together, the absence of cytokine-induced immune responses in the gastrointestinal tract is in alignment with the absence of strong histopathological changes in the same tissue.

## 4. Conclusions

To summarize, our study within a restrictive feeding exposure regime for six weeks showed direct effects of DON on protein gain in rainbow trout regardless of the source of industrially relevant DON levels (natural: 1200 or pure: 1600 µg/kg), below the current European Commission recommendation limit of 5000 µg/kg. Moreover, we revealed time-related DON effects on liver histological parameters, an early response at week 1 and a recovery by week 6. Apparently, rainbow trout exposed restrictively to a certain amount of naturally contaminated DON diet (1200 µg/kg) develop an adaptation mechanism to recover overtime and eliminate necrosis and haemorrhage in the liver. The adaptation process is not fully understood, and therefore further research is recommended on the antioxidant system, detoxification capacity, and toxicokinetics of DON in trout. Moreover, additional research is required in order to shed light on the potential combined effects of *Fusarium* toxins since we observed adaptation over time within the natural DON group. The *ad libitum* access to DON after the restrictive exposure did not impair feed intake but suppressed growth performance. At the end of the *ad libitum* exposure, histopathological damage was detected in the liver but not in the gastrointestinal tract, where no immunomodulating properties of DON were present. The severity of DON that was described during *ad libitum* exposure is an outcome of the absolute amount of DON ingested that led to a threshold dose that trout could not further tolerate. Finally, because DON severity might be species-specific, similar in vivo investigations are also recommended in other farmed fish species.

## 5. Materials and Methods

This study (project number AVD2330020198084) was carried out in accordance with the Dutch law on the use of animals (Act on Animal Experiments) for scientific purposes, and it was approved by the Central Committee on Animal Experiments (CCD) of The Netherlands. The experiment was executed at the experimental facilities of the Alltech Coppens Aqua Centre (Leende, The Netherlands).

### 5.1. In Vivo Experimental Procedure

Rainbow trout (*Oncorhynchus mykiss*) were obtained from a commercial trout farm (Mohnen Aquaculture GmbH, Stolberg, Germany) and kept in a recirculating aquaculture system (RAS). Fish with an average weight of 8 g were randomly distributed over 15 tanks with 30 fish per tank. All five treatments (CON, ND1, ND2, PD1, PD2) were tested in triplicate (3 tanks per treatment). Each tank was additionally aerated and had a volume of 120 L. A cooling system maintained the water temperature constant at 14 ± 0.5 °C, and a photoperiod of 17 h light and 7 h dark was applied during the experiment. Water physiochemical parameters were monitored and maintained within the allowed levels: pH: 7.0–8.5, NH_4_^+^: <1 mg/L, NO_2_^−^: <0.5 mg/L, alkalinity: 2.0–5.0, and oxygen (O_2_): 8 mg/L. Fish were daily checked for signs of abnormal behaviour (e.g., cannibalism, irregular swimming patterns, lethargic and weak individuals hiding at the bottom of the tank for a while), diseases, wounds, and mortalities.

The experiment lasted eight weeks and consisted of two periods: a 6-week restricted feeding period followed by a 2-week *ad libitum* feeding period. Before the start of the experiment, all fish were acclimated to the facilities for a week and fed a standard commercial trout diet. To assess the direct effect of DON, fish were fed restrictively for six weeks according to their metabolic body weight (12 g/kg^0.8^/d) by handfeeding twice per day to ensure fixed daily intakes of DON and an equal amount of feed given to each tank. To evaluate the effect of DON in combination with impacts on feed intake, during the last two weeks of the experiment, fish were fed *ad libitum* twice daily for one hour. Fish had reached satiation when uneaten pellets remained on the bottom of the tank or floating on the water’s surface for more than 10 min or when the feeding time of one hour was over. Uneaten pellets were removed by siphoning and counted to determine feed intake.

The day before the start of the experiment (before distribution to tanks), from the initial population, *n* = 6 fish were euthanized for tissue sampling and *n* = 20 fish for determining the initial body composition at time point zero. At the start and at the end of both feeding periods (i.e., weeks 6 and 8), fish were weighed per tank and counted for the calculation of performance parameters. At the end of the restrictive feeding period (week 6), *n* = 5 fish per tank were euthanized and stored at −20 °C for determination of body composition. After one week of restrictive feeding (week 1), at the end of the restrictive feeding period (week 6), and at the end of the *ad libitum* feeding period (week 8), liver and tissue samples from the gastrointestinal tract (pyloric caeca, midgut, and hindgut) were collected from *n* = 2 fish per tank and stored for histopathological examination and gene expression analysis. Additionally, total liver weight and total body length were recorded for all fish sampled for tissues. Overall, handling of the fish was avoided as much as possible, and the fish were euthanized by an overdose of benzocaine (dissolved in water at 0.5 mL/L).

### 5.2. Experimental Diets

Five experimental diets were formulated, of which one was the control diet (CON), which aimed to have as little DON content as possible, and four diets had different concentrations and origins of DON included. Effects induced by natural DON originating from a batch of “contaminated” wheat (further information below) were compared to effects induced by pure DON. The pure DON was produced by extracting and purifying it from a fermentation medium of *Fusarium graminearum*, purchased from Fermentek Ltd. (Jerusalem, Israel). The two DON concentrations of 800 and 1600 µg/kg are anticipated to be below the threshold level for DON (<5000 µg/kg) advised by the European Commission [37].

Before feed production, various batches of wheat were analysed for DON to find two batches of wheat: a “clean” for the control diet and a “contaminated” batch. The “contaminated” batch was designed to have the highest possible DON content. DON was quantified by liquid chromatography/tandem mass spectrometry (LC-MS/MS) at the Alltech 37+ mycotoxin laboratory (ISO/IEC 17025:2005 accredited), Dunboyne, Ireland. Next to DON, the “contaminated” DON wheat also contained small amounts of other mycotoxins. The analysed contents in the “contaminated” wheat was for DON 3842 µg/kg, for DON-3-Glucoside (DON3Glc) 124 µg/kg, for *Fusarenon* X (FX) 29 µg/kg and Alternariol 8 µg/kg. 

In order to compose a control diet with minimal dietary contamination of mycotoxins, the CON diet was fully fishmeal and fish oil-based with the inclusion of 40% of the “clean” wheat source (Table 4). To compose experimental diets with natural DON concentrations of 800 µg/kg (ND1 diet) and 1600 µg/kg (ND2 diet) the “clean” wheat was partially (ND1) or fully (ND2) exchanged for the “contaminated” wheat. To compose experimental diets with pure DON concentrations of 800 µg/kg (PD1 diet) and 1600 µg/kg (PD2 diet), the CON diet was supplemented with the appropriate amount of pure DON. This approach resulted in five experimental diets, all isonitrogenous and isoenergetic. The experimental diets were produced as 2 mm extruded pellets by Research Diet Services (Wijk bij Duurstede, The Netherlands). Following pelleting, diets were analysed for mycotoxin content to confirm the anticipated DON contamination levels. The results confirmed the low occurrence of DON (70 µg/kg) and other toxins in the control diet (Table 4). DON levels in the naturally contaminated diets, ND1 and ND2, were slightly lower than anticipated at ~700 and ~1200 µg/kg, respectively, and also contained small amounts of Enniatin A/A1 and Enniatin B/B1. DON concentrations in PD1 and PD2 diets were close to the anticipated levels at ~800 and ~1600 µg/kg, respectively (Table 4).

### 5.3. Chemical Analysis of Feeds and Fish

Feed samples were analysed for: dry matter (DM) content by drying at 103 °C until constant weight for 4 and 24 h, respectively (ISO 6496, 1999), crude protein (CP) based on nitrogen × 6.25 using the Kjeldahl method (ISO 5983, 2005), fat after an initial acid-hydrolysis step followed by a petroleum-diethyl ether extraction (ISO 6492, 1999), ash content after incineration at 550 °C for 4 h (ISO 5984, 2002), and gross energy (GE) content with the adiabatic bomb calorimeter method (ISO 9831, 1998). Fish carcass samples were analysed with the same methods for CP and GE. All chemical analyses were performed by Nutricontrol (Veghel, The Netherlands).

### 5.4. Histopathological Examination of Liver and Gastrointestinal Tract

At the end of weeks 1, 6, and 8, *n* = 2 fish per tank (i.e., six per treatment) were sampled for histopathological assessment of the liver and gastrointestinal tract (and *n* = 6 from the initial population before the experiment starts). Although, due to time constraints, only the diets with the highest DON doses (ND2 and PD2) were further analysed and compared with the CON diet. Overall, two pieces from each liver and a piece from each part of the gastrointestinal tract (pyloric caeca, midgut, and hindgut) were placed into embedding cassettes and fixed by immersion in 10% buffered formaldehyde for three days at room temperature. Samples were later transferred to 70% ethanol until dehydration and embedded in paraffin wax according to standard histological procedures. All liver and intestinal tissue blocks were cut into 5 μm thick paraffin sections, mounted onto microscope slides, and stored in an oven at least overnight, followed by staining (details are described below). Pictures were captured with a Leica DM6 microscope (Leica Microsystems, Wetzlar, Germany).

Liver sections were stained with Periodic acid-Schiff’s (PAS) reagent to distinguish between lipid- and glycogen-type vacuoles, followed by staining with Crossman’s trichrome (Mason) for coloration of connective tissue (collagen). Liver sections were also stained with Haematoxylin and Eosin (H and E) to assess cellular and nuclear morphology. Glycogen accumulation in the hepatocytes was observed as pink-purple areas of PAS-positive material, while lipid accumulation was observed as well-defined white spherical droplets. Glycogen and lipid vacuolisation were scored as follows: low (1) moderate (2), and high (3). PAS-Crossman-stained liver sections were also screened for histopathological aberrations, including signs of haemorrhage and inflammation, the latter identified as infiltrates of nucleated leukocytes, by scoring “Yes” or “No”. We categorised and scored necrotic presence as follows: no necrosis (0), mild (1), moderate (2), and severe (3). Liver sections stained with H&E were used to assess nuclear morphology as follows: presence (“Yes”) or absence (“No”) of “pyknotic”, “dislocated”, and “pleomorphic” nuclei. All parameters were assessed for 10 single random frames per sampled fish (*n* = 5 from each liver piece) stained with PAS (× 20 magnification) and H&E (× 10 magnification). Finally, to exclude bias, blind histological assessment of liver samples was carried out by two evaluators. When scores were not in agreement, differences were discussed until consensus was reached.

Gastrointestinal tract sections were stained with Alcian blue (pH 2.5), a stain that is used to visualise acidic epithelial and connective tissue mucins, followed by Crossman to enhance the contrast between goblet cells (GC) and supranuclear vacuoles (SNV). Alcian blue staining revealed a heterogeneous population of mucus-producing cells, identified by a range of blue stain intensity, presumably because GC which secrete a combination of acidic and neutral mucus would be visible as dark blue and GC which secrete acidic mucus would be visible as light blue. All the GC which stained blue were counted, regardless of intensity, around the perimeter of each mucosal fold (MF) and expressed as the number of GC per µm^2^ of MF. Eosinophilic granulocytes (EG), if present, were counted as cells with light pink coloured cytoplasm. 

For evaluating histological parameters in each section of the gastrointestinal tract, we randomly picked *n* = 10 well-oriented (simple) fold units and measured the following parameters: (a) thickness of sub-epithelium mucosa (SM), measured as the distance between the point where neighbouring folds lose contact with each other prior to the collagenous (greenish-blue) layer of connective tissue (b) stratum compactum height (SC); (c) mucosal fold height (MFH); (d) mucosal fold width (MFW); (e) average lamina propria width (LP) from three different areas; (f) average supranuclear vacuoles width (SNV) from two sides; (g) enterocytes width (EW) calculating from MFH, LP and SNV; (h) stratum granulosum height (SG), defined as the layer bordered by SC and muscular layer, (i) muscularis (MS); and (j) MS (consisting of the inner circular (cm) and longitudinal (lm) layer) was determined as the layer between SG and the thin outermost layer of connective tissue, (k) serosa (SE). Pictures were imported into the ImageJ software (version 1.53 q [38]), and all the above-mentioned histological parameters were measured with the ROI manager function. Our scoring system is an updated quantitative approach based on previously used parameters to semi-quantitatively score soybean-induced enteritis in Atlantic salmon and common carp [39,40]. Finally, an example of the measurements on the described parameters (a–k) to evaluate the effects of DON along the gastrointestinal tract (pyloric caeca, midgut, and hindgut) of trout is available in Appendix A.

### 5.5. Gene Expression

Tissue samples from the pyloric caeca and hindgut of rainbow trout (*n* = 2 per tank, i.e., 6 per treatment), fed the experimental diets CON, ND2, and PD2, were analysed for gene expression analysis at the end of the experiment (week 8). Small (2 mm in size) tissue samples were placed in Eppendorf tubes filled with RNA*later*, stored at room temperature overnight, and then transferred to −20 °C until RNA extraction. Total RNA was extracted using the RNeasy^®^ mini kit (Qiagen), including on-column DNase treatment with a RNase-free DNase set (Qiagen), according to the manufacturer’s instructions. Total RNA was stored at −80 °C until use. Before cDNA synthesis, 1 µg RNA was treated with DNase I, Amplification Grade (InvivoGen). cDNA was synthesised using random primers (300 ng) and Superscript III First-Strand Synthesis for RT-PCR following the manufacturer’s (InvivoGen) protocol. cDNA samples were diluted (1:20) in nuclease-free water and used for real-time quantitative PCR (RT-qPCR) with ABsolute QPCR, SYBR Green Mix (Thermo Fisher Scientific) in a Rotor-Gene 6000 (Corbett Research). Fluorescence data were retrieved and analysed by Rotor-Gene Q Series software (version 2.1.0 Build 9). Gene expression was measured as a relative expression ratio calculated according to the Pfaffl method [41]. Take-off values of experimental samples were calibrated against a common reference (calibrator) and normalised against the reference gene elongation factor (ELF-1α) of rainbow trout. Specific primer sequences for the reference gene and interleukin-1β (IL-1β), interleukin-8 (IL-8), copies of tumor necrosis factor-α (TNF-α andTNF-α3) are in Table 5. Primer pairs had been validated as gene copy-specific by sequencing of PCR products prior to this analysis. The following PCR reaction conditions were applied: 95 °C for 15 min, followed by 35 cycles of 95 °C for 20 s, 60 °C for 20 s, and 72 °C for 20 s. To ensure the specificity of amplification, a melting curve analysis was performed with a hold of 60 °C for 1 min and a melting curve temperature ranging from 60 °C to 99 °C with a gradual increase of 0.5 °C every 5 s. 

### 5.6. Calculations

The average initial body weight (IBW, g) and the final body weight (FBW, g) per fish were determined by batch-weighing of the tank biomass and dividing by the number of individual fish. Feed intake (FI) was defined as the average amount of feed (g) consumed by a fish, converted based on the DM content of the feed (g/kg). 

By using the following formulas, we calculated per feeding period (restricted and *ad libitum*):Weight gain (g) = FBW − IBW(1)
Growth (g/d) = weight gain/days(2)
Specific growth rate (SGR, %/d) = ((ln FBW − ln IBW)/days) × 100(3)
Feed conversion ratio (FCR) on DM basis = FI/Weight gain(4)
Hepatosomatic index (HSI, %) = (liver weight/W) × 100(5)
Condition factor (K) = (W/L^3^) × 100(6)
where W is the individual FBW of the tissue sampled fish and L its body length (cm).
Retained protein (g/fish) = FBW × FPC − IBW × IPC(7)
where FPC is the protein content (g) in the fish body at the end and IPC the protein content (g) at the start.
Protein retention efficiency (%) = (Retained protein/CPI) × 100(8)
where CPI is the dietary protein intake (g/fish) calculated as average FI of an individual x protein content in the feed.

Similarly for Retained energy (MJ/fish) and energy retention efficiency (%)
Retained energy = FBW × FEC − IBW × IEC(9)
where FEC is the gross energy content (MJ) in the fish body and the end and IEC the protein content (g) at the start.
Energy retention efficiency (%) = (Retained energy/GEI) × 100(10)
where GEI is the dietary gross energy intake (MJ/fish) calculated as average FI of an individual x gross energy in the feed.

### 5.7. Statistical Analysis

For growth parameters (nitrogen and energy retention efficiencies), the experimental unit was the tank, and data were expressed as a treatment mean derived from the three replicates. A one-way analysis of variance ANOVA using a general linear model (GLM) was used to evaluate the effect of dietary DON on the dependent variables. HSI, condition factor, and gene expression measurements were performed on individual fish (means derived from six replicates per treatment, two per tank), therefore a generalised linear mixed model was applied with the tank used as a random effect. However, the tank effect was not significant (*p* > 0.05) and therefore not included in the results. When a significant difference was found (*p* ≤ 0.05), a Tukey’s honestly significant difference (HSD) *post hoc* test with multiple comparisons (95% level of significance) was used to compare treatment means. 

For the outcome variables in the gastrointestinal tract and continuous scores in the liver (glycogen and lipid vacuolisation score, and necrosis score), a general linear regression was performed with diet (CON, ND2, PD2) and time (weeks 1, 6, and 8), and their interaction weas included in the model. Before the statistical analysis, the 10 measurements (liver areas and intestinal folds) were averaged per fish. Gastrointestinal tract parameters were analysed separately per part of the intestine (pyloric caeca, midgut, and hindgut). The model residuals were considered normal when skewness and kurtosis were between −2 and 2. The scores were expressed as least square means (*n* = 54, 6 per diet per time point). For the yes/no liver data (nuclei pyknosis and pleomorphism, necrosis, haemorrhage, inflammation), a logistic regression analysis was performed, which included diet, time, and their interaction in the model. The scores were expressed as frequencies (%) (*n* = 540, 60 per diet per time point). As the 10 measurements within a fish are not independent, a random fish effect was included using the exchangeable correlation structure (GEE model). A marginal R^2^ for the GEE model was calculated, which is interpreted similarly to the R^2^ in ordinary least square regression models [42]. 

The IBM Statistical Package for the Social Sciences (SPSS) programme (v 23.0; New York, NY, USA) was used to perform the statistical analyses for growth performance and gene expression analysis. Histological data were analysed with SAS software^®^ (version 9.4, SAS Institute, Cary, NC, USA).

## Figures and Tables

**Figure 1 toxins-14-00810-f001:**
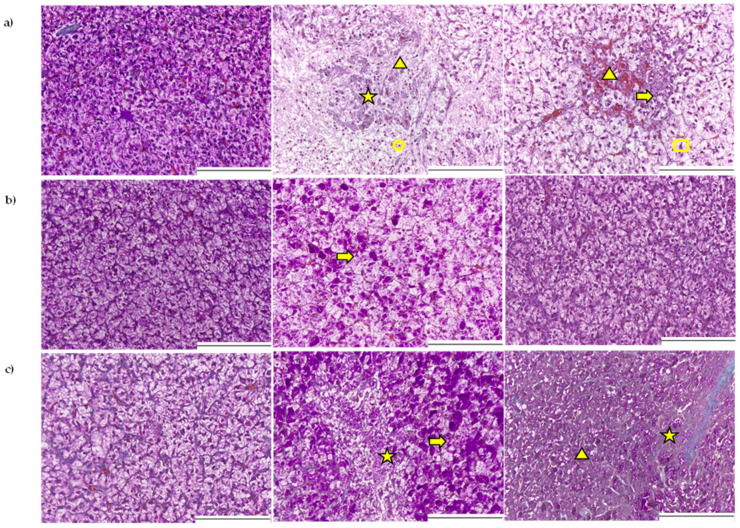
Representative examples of histological sections of the liver from rainbow trout fed diets of control (CON), natural (ND2), and pure DON (PD2), restrictively for one week (**a**), six weeks (**b**), and *ad libitum* for two weeks (**c**). The yellow arrows indicate profound infiltration of presumed *leucocytes*; stars highlight necrotic areas; triangles show the presence of a haemorrhage; circles indicate pyknotic nuclei; and squares indicate pleomorphic nuclei. Staining: PAS-Crossman; Magnification: ×20; White scale bar = 200 µm.

**Figure 2 toxins-14-00810-f002:**
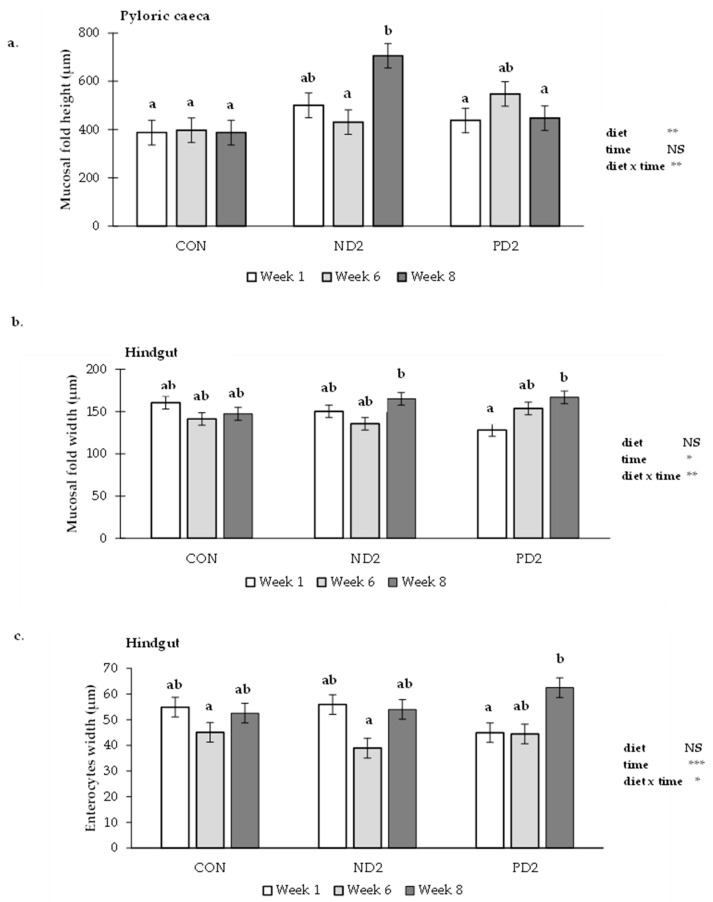
Interaction effects between the experimental diets and exposure time on the (**a**) mucosal fold height in the pyloric caeca, (**b**) mucosal fold width in the hindgut, and (**c**) enterocyte width in the hindgut of rainbow trout. Experimental diets refer to control (CON), natural DON (ND2), and pure DON (PD2), and time to restrictive exposure for 6 days (week 1), 40 days (week 6), and ad libitum exposure for 15 days (week 8). The error bars indicate the standard error of the mean; NS: not significant, *: *p* ≤ 0.05, **: *p* ≤ 0.01, ***: *p* ≤ 0.001. Treatments lacking a common letter (a, b) are statistically different (*p* ≤ 0.05) according to Tukey’s multiple comparison test.

**Figure 3 toxins-14-00810-f003:**
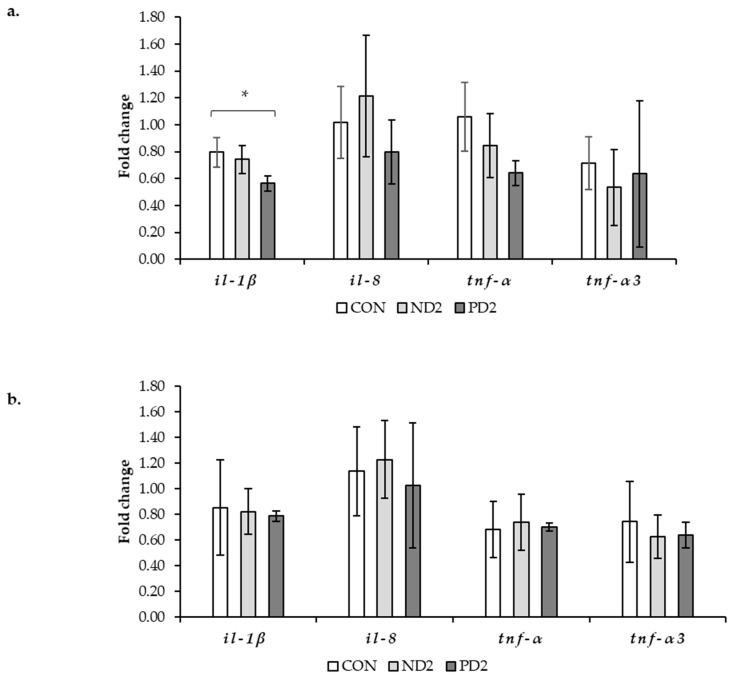
Relative gene expression (fold change) of pro-inflammatory cytokines interleukin-1*β*, interleukin-*8*, and two copies of tumor necrosis factor-*α* at the end of the experimental period (week 8) in (**a**) the pyloric caeca and (**b**) the hindgut of rainbow trout fed the experimental diets: control (CON), natural (ND2), and pure DON (PD2). The number of records used in the statistical analysis per gene ranged from 13 to 14. Primer pairs were gene copy specific; accession numbers of the gene variants amplified can be found in Table 5. * refers to a significant difference compared to the CON diet (*p* ≤ 0.05).

**Table 1 toxins-14-00810-t001:** Effects of DON on the performance of rainbow trout fed the experimental diets: control (CON; DON < 100 µg/kg), naturally DON-contaminated diets (ND1; DON = 700 µg/kg and ND2; DON = 1200 µg/kg), and pure DON-contaminated diets (PD1; DON = 800 µg/kg and PD2; DON = 1600 µg/kg) during a 6-week restrictive feeding period.

	Experimental Diets						
Performance Parameters	CON	ND1	ND2	PD1	PD2	SEM	*p*-Value
Initial BW (g)	8.0	8.0	7.4	8.2	8.1	0.21	NS
Final BW (g)	36.5	36.1	35.5	36.8	35.5	0.31	NS
Growth (g/d)	0.71	0.70	0.70	0.72	0.68	0.008	NS
SGR (% BW/d)	3.80	3.78	3.91	3.76	3.69	0.069	NS
FCR	0.68	0.70	0.70	0.69	0.71	0.008	NS
HSI (%)	4.4	4.4	4.7	4.2	3.8	0.36	NS
Condition factor (K)	2.1	2.0	1.8	2.0	1.9	0.09	NS
Retained protein (g/fish)	4.2 ^b^	4.0 ^ab^	3.9 ^a^	4.1 ^b^	3.8 ^a^	0.05	**
Protein retention efficiency (%)	51.0 ^c^	49.1 ^ac^	48.1 ^ab^	50.4 ^bc^	47.1 ^a^	0.59	**
Retained energy (MJ/fish)	0.19	0.19	0.19	0.19	0.19	0.002	NS
Energy retention efficiency (%)	45.9	45.8	44.3	45.8	44.6	0.58	NS

BW: body weight, SGR: specific growth rate, FCR: feed conversion ratio on dry matter basis, HSI: hepatosomatic index, SEM: standard error of means, NS: not significant, **: *p* ≤ 0.01, values in the row with different superscripts (a, b, c) are significantly different (*p* ≤ 0.05) according to Tukey’s multiple comparison test.

**Table 2 toxins-14-00810-t002:** Effects of DON on the performance and feed intake capacity of rainbow trout fed the experimental diets: control (CON; DON < 100 µg/kg), naturally DON-contaminated diets (ND1; DON = 700 µg/kg and ND2; DON = 1200 µg/kg), and pure DON-contaminated diets (PD1; DON = 800 µg/kg and PD2; DON = 1600 µg/kg) during a 2-week *ad libitum* feeding period.

	Experimental Diets						
Performance Parameters	CON	ND1	ND2	PD1	PD2	SEM	*p*-Value
Initial BW (g)	36.6	36.1	35.5	36.6	36.1	0.47	NS
Final BW (g)	67.8 ^a^	64.2 ^ab^	61.9 ^b^	65.1 ^ab^	63.1 ^ab^	1.06	*
Feed intake (g/fish/d)	1.71	1.72	1.58	1.71	1.67	0.045	NS
Feed intake (g/kg^0.8^/d)	18.9	19.5	18.9	19.2	19.2	0.48	NS
Growth (g/d)	2.09 ^a^	1.88 ^ab^	1.76 ^b^	1.90 ^ab^	1.80 ^b^	0.048	**
SGR (% BW/d)	4.12 ^a^	3.84 ^ab^	3.70 ^b^	3.83 ^ab^	3.73 ^b^	0.064	**
FCR	0.79 ^a^	0.86 ^ab^	0.85 ^ab^	0.84 ^ab^	0.89 ^b^	0.017	*
HSI (%)	3.9	3.4	3.8	3.0	3.0	0.40	NS
Condition factor (K)	2.0	1.9	1.9	1.9	2.0	0.07	NS

BW: body weight, SGR: specific growth rate, FCR: feed conversion ratio on dry matter basis, HSI: hepatosomatic index, SEM: standard error of means, ns: not significant, *: *p* ≤ 0.05, **: *p* ≤ 0.01, values in the row with different superscripts (a, b) are significantly different (*p* ≤ 0.05) according to Tukey’s multiple comparison test.

**Table 3 toxins-14-00810-t003:** Pathological indicators observed in trout livers with histological assessment after both restricted (week 1 and week 6) and *ad libitum* (week 8) feeding of the experimental diets: control (CON), natural DON (ND2), and pure DON (PD2).

Pathological Indicators		Experimental Diets				*p*-Value			
	+Week	CON	ND2	PD2	SEM	Diet	Time	Diet × Time	R^2^ (Fish Effect) ^‡^
Vacuolization Score									
Glycogen ^#^	1	2.5	2.3	1.8 ^y^					
	6	2.0	1.9	1.9 ^y^	0.19	***	NS	*	0.42 (NA)
	8	2.4 ^b^	2.0 ^b^	1.0 ^a,x^					
Lipid ^#^	1	1.6	1.8	1.9					
	6	2.0	1.8	1.9	0.13	NS	NS	NS	0.16 (NA)
	8	1.9	2.0	2.0					
Nuclei characteristics ++									
Pyknotic (%)	1	55 ^a^	95 ^b^	93 ^b^					
	6	33	87	78		***	NS	NS	0.36 (0.50)
	8	23 ^a^	72 ^b^	15 ^a^					
Pleomorphic (%)	1	35 ^a^	82 ^b^	82 ^b^					
	6	25	65	68		*	NS	NS	0.29 (0.51)
	8	18 ^a^	83 ^b^	17 ^a^					
Other indicators									
Necrosis (%)	1	28	62 ^y^	37 ^x^					
	6	25	22 ^x^	39 ^x^		NS	NS	**	0.21 (0.22)
	8	18 ^a^	45 ^a, x, y^	93 ^b, y^					
Necrosis score ^#^	1	0.3	1.2	0.5 ^x^					
	6	0.4	0.5	0.5 ^x^	0.20	*	NS	*	0.35 (NA)
	8	0.2 ^a^	0.7 ^ab^	1.1 ^b y^					
Haemorrhage (%)	1	12 ^a^	50 ^b y^	13 ^a^					
	6	5	5 ^x^	12		***	NS	***	0.15 (0.04)
	8	3 ^a^	15 ^b x^	30 ^b^					
Inflammation (%)	1	0 ^a^	2 ^a x^	13 ^b^					
	6	3 ^a^	45 ^b y^	9 ^a^		*	NS	*	0.17 (0.16)
	8	5	22 ^y^	5					

+ Liver samples collected before the start of the experiment (Week 0) were homogeneously characterized with low glycogen and lipid vacuolization (score 1), without nuclei alternations (pyknosis and pleomorphism) and pathological findings (necrosis, haemorrhage, inflammation). ++ Scores of pyknosis and pleomorphism were used as indicators to evaluate cell viability. ^#^ Glycogen, lipid vacuolisation, and necrosis scores were analysed with a general linear model (*n* = 54). The other pathological indicators were analysed with a logistic regression model with fish as a random effect (*n* = 540). ^‡^ The fish effect is the proportion of all unexplained variation due to fish. This was only estimated for pathological indicators analysed by logistic regression. The R^2^ is the proportion of variance explained by the model. Standard error of the mean: SEM, Not significant: NS, *p* ≤ 0.05: *, *p* ≤ 0.01: **, *p* ≤ 0.001: ***, Not applicable: NA, a, b: different superscripts within rows mean differences between treatments within a week with a *p* ≤ 0.05. x, y: different superscripts within columns mean differences between weeks within a treatment with a *p* ≤ 0.05.

**Table 4 toxins-14-00810-t004:** Ingredient composition, proximate, and mycotoxin analysis of the experimental diets: control (CON), naturally DON-contaminated diets (ND1 and ND2), and pure DON-contaminated diets (PD1 and PD2).

			Experimental Diets		
Ingredient (%)	CON	ND1	ND2	PD1	PD2
Wheat (no DON)	40.00	18.00	-	40.00	40.00
Wheat (DON contaminated)	-	22.00	40.00	-	-
Pure DON	-	-	-	0.00009	0.00016
LT fishmeal	49.02	49.02	49.02	49.02	49.02
Fish oil	9.90	9.90	9.90	9.90	9.90
Mineral and vitamin premix ^1^	1.08	1.08	1.08	1.08	1.08
Analysed nutrient composition ^2^ (%)					
Dry Matter	96.5	94.3	94.0	93.3	95.7
Protein	41.8	41.5	41.3	41.7	41.6
Fat	15.8	16.3	16.2	16.3	16.2
Ash	9.7	9.9	9.8	9.8	9.5
Gross Energy (MJ/kg)	21.62	21.60	21.64	21.62	21.57
Mycotoxins concentration (µg/kg) ^2^					
DON ^3^	70	679	1192	781	1566
Enniatin A/A1 ^4^	1.2	12.1	12.9	-	-
Enniatin B/B1 ^4^	-	8.6	19.9	-	-
T2 Toxin	-	-	3.8	-	-
Ergotamin(in)e	2.7	3.4	-	2.1	-
Ergocryptin(in)e	4.1	-	-	-	-

^1^ Commercial premix from Alltech Coppens that meets NRC, 2011 requirements for rainbow trout. ^2^ On dry matter basis. ^3^ In the main text, the rounded levels of DON are mentioned: ND1: 700, ND2:1200, PD1: 800 and PD2: 1600 µg/kg. ^4^ Wheat batches were not screened for Enniatin A/A14 and Enniatin B/B14.

**Table 5 toxins-14-00810-t005:** Summary of selected genes primer sequences used for RT-qPCR.

Target Gene	Accession Number	Forward Primer Sequence 5’-3’	Reverse Primer Sequence 5’-3’
elf-1α	AF498320.1	TCTACAAAATCGGCGGTA	CCTCAGTGGTGACATTAGC
il-1β	AJ557021.2	CACCACCACCACCAAT	AAGAGGAAGCGAACCG
il-8 ^+^	NM001124362.1	TGTCAGCCAGCCTTG	ACATCCAGACAAATCTCCT
tnf-α	AJ277604.2	GGCTGTGTGGGGTC	GCTTCAATGTATGGTGGG
tnf-α3	HE798146.1	TACCAAGAAACAAGATCACA	TCTGTCCACTCCACTGA

+ there exist 2–3 paralogs for IL-8 with minor nucleotide differences, the primers were chosen to amplify all paralogs.

## Data Availability

The corresponding author can be contacted if access to the data is desired.

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
