# Peer review of "Time- and Dose-Dependent Effects of Dietary Deoxynivalenol (DON) in Rainbow Trout (Oncorhynchus mykiss) at Organism and Tissue Level"

_toxins, 2022, doi:10.3390/toxins14110810_

Round 1
Reviewer 1 Report
This study is dedicated to very important problem associated with mycotoxin, especially DON prevalence in aquaculture fish feed and related threats to fish quality and health issues.
The authors have provided interesting evaluation of the dietary exposure to naturally contaminated and artificially added DON levels to fish (juvenile rainbow trout) feed and tested 8 weeks the effect of restrictive (6 weeks) and libitum feeding.
Comparing to the previous study, authors provide some clearance of the differences and dose dependent effects, whereas the data indicate directly the liver damage at the increased concentrations of DON and prolonged exposure to contaminated feed.
It would be interesting if authors could also indicate to the issues of multi-mycotoxin contamination, especially taking into account T-2 and other more toxic trichothecene mycotoxins. From the author studies it seems that there are plans to continue this study.
it should be noted that all the methods are clear, the experiments were provided according to standartized practices.
It would be interesting, if authors could also to indicate whether these effects could be predicted also for other aquaculture farmed fish species.
Otherwise it is recommended to publish this paper in the present form.
Author Response
Please find our responses in the attached document.

Reviewer 2 Report
A very interesting manuscript with a topic that is increasingly important due to the increasing presence of mycotoxins in animal feed.
All potential doubts regarding the Manuscript and the planning of the research were emphasized by the authors themselves. Therefore, I have no objections to the Manuscript itself.
This especially applies to the experimental period, which was too short for any major differences between the groups, and the share of DON in the highest doses was significantly lower than legally prescribed in the EU. The authors noticed all this and left the possibility for future research.
Therefore, I am very satisfied to recommend this Manuscript for publication.
(The only necessary correction refers to line 278, where there was a typographical error and an unnecessary change of font to Italic in one part of the sentence.)
Author Response

(The authors gave the same response as above.)

Reviewer 3 Report
The manuscript entitled Time- and dose-dependent effects of dietary deoxynivalenol 2 (DON) in rainbow trout (Oncorhynchus mykiss) at organism and 3 tissue levels, investigated how different concentrations of DON can alter the histopathological signature in GI of trout. Furthermore, the authors performed a phenotypic assessment after exposure and measured gene expressions for some of the inflammatory genes.
although the manuscript was written well and the experiments were designed nicely, the manuscript lacks novelty. The essays included were limited. for instance, lots of other genes that are more relevant than inflammatory genes can be included in the analyses. furthermore, other biochemical studies like antioxidants and many other markers are missing. Tissue uptake analyses are needed.
minor comments.
L 450, any definition for abnormal behavior?
L 537, what kind of staining, H and E?
L631, I recommend running the statistical analyses using a mixed model instead of ANOVA. This will allow you to include the tank effect.
Author Response

(The authors gave the same response as above.)

Round 2
Reviewer 3 Report
authors provided answers for the minor comments. they haven't commented on my primary concern. I'm afraid that the MS still lacks novelty. this is a nice study but the authors might consider more analyses and publish it in the next run.